# Saliency Detection with Moving Camera via Background Model Completion

**DOI:** 10.3390/s21248374

**Published:** 2021-12-15

**Authors:** Yu-Pei Zhang, Kwok-Leung Chan

**Affiliations:** Department of Electrical Engineering, City University of Hong Kong, Hong Kong, China; ypzhang5-c@my.cityu.edu.hk

**Keywords:** background modeling, background subtraction, foreground segmentation, saliency detection, PTZ camera, mobile camera

## Abstract

Detecting saliency in videos is a fundamental step in many computer vision systems. Saliency is the significant target(s) in the video. The object of interest is further analyzed for high-level applications. The segregation of saliency and the background can be made if they exhibit different visual cues. Therefore, saliency detection is often formulated as background subtraction. However, saliency detection is challenging. For instance, dynamic background can result in false positive errors. In another scenario, camouflage will result in false negative errors. With moving cameras, the captured scenes are even more complicated to handle. We propose a new framework, called saliency detection via background model completion (SD-BMC), that comprises a background modeler and a deep learning background/foreground segmentation network. The background modeler generates an initial clean background image from a short image sequence. Based on the idea of video completion, a good background frame can be synthesized with the co-existence of changing background and moving objects. We adopt the background/foreground segmenter, which was pre-trained with a specific video dataset. It can also detect saliency in unseen videos. The background modeler can adjust the background image dynamically when the background/foreground segmenter output deteriorates during processing a long video. To the best of our knowledge, our framework is the first one to adopt video completion for background modeling and saliency detection in videos captured by moving cameras. The F-measure results, obtained from the pan-tilt-zoom (PTZ) videos, show that our proposed framework outperforms some deep learning-based background subtraction models by 11% or more. With more challenging videos, our framework also outperforms many high-ranking background subtraction methods by more than 3%.

## 1. Introduction

High-level applications such as human motion analysis [1] and intelligent transportation system [2] demand the localization of targets in the video. For instance, in video surveillance, humans are detected for motion recognition. Vehicles are located in an intelligent transportation system. This foremost task can be achieved via saliency detection. Assuming that the background scene possesses invariant characteristics, a target is detected due to its deviated visual cues. One approach is to formulate the task as background/foreground segmentation. With the estimated background scene model, the foreground (i.e., saliency) is segmented by a pixelwise background subtraction algorithm. The foreground is defined as a region of interest that is not a part of the background. While it is usually a moving object, it may also stop suddenly. Therefore, the region outside the foreground such as the shadow is considered as the background. The background may have moving elements, but their motions are not of interest.

However, the two assumptions—invariant background and deviation of foreground—may be violated in some circumstances. For instances, background motion and illumination change can result in false detection (false positive). With the pre-generated background model, background pixels may be predicted as foreground pixels. This is a false positive (FP) error that is due to image feature(s) of the background pixel different from the background model. On the other hand, camouflage and intermittent object motion will result in missing detection. This false negative (FN) error is due to the fact that some foreground pixels may be erroneously identified as the background pixels if they have similar image feature(s) to the background model.

The saliency detection task becomes more challenging with the use of moving camera. The assumption of a static background is violated. Videos can be captured by a pan-tilt-zoom (PTZ) camera or free-moving (e.g., hand-held) camera. The pan and tilt camera motions can capture a broader vision field. Zooming can focus the region of interest with a higher resolution. While these camera motions are constrained, free-moving camera can perform any kind of motion without constraint. Systems that can handle such type of video are of interest. They demand sophisticated techniques for generating and maintaining the background model, as well as foreground segmentation.

Researchers have proposed various background subtraction algorithms. Many background subtraction methods are deterministic, i.e., background/foreground segmentation is achieved based on hand-crafted features. One earliest approach is to adopt statistical models [3,4]. Elgammal et al. [5] utilized kernel estimator to characterize the probability density function (pdf) of the background pixels. Some researchers have presented the survey on the background subtraction techniques [6,7]. Sobral and Vacavant [8] evaluated 29 background subtraction methods. In general, background subtraction comprises three main parts—background modeler, background/foreground classifier, and background updating.

Another approach is to use neural networks for saliency detection. Its cognitive power is made possible with the structure simulating complex connectivity of neurons. Maddalena and Petrosino [9] proposed Self Organizing Background Subtraction (SOBS). The background scene is modeled with the weights of the neurons. The network compares the current image frame with the background model and outputs pixelwise background/foreground classification. Recently, a popular approach is to develop deep learning models, such as convolutional neural network (CNN). The layered structure can accommodate multi-scale representation, with which image data are transformed and abstract features are extracted. Wang et al. [10] proposed a basic CNN model, with which multi-resolution CNN and cascaded CNN architectures were designed for object segmentation. Lim and Keles [11] proposed an encoder–decoder network for object segmentation. The encoder part is a triple CNN for multi-scale feature extraction. The concatenated feature map is fed to a transposed convolutional network in the decoder part. They [12] further proposed another model that uses feature pooling module on top of the encoder part.

In this paper, we propose a new framework, called saliency detection via background model completion (SD-BMC), that comprises a background modeler and a deep learning background/foreground segmentation network. Our framework can detect saliency in videos captured by moving camera. The results, obtained from the benchmark datasets, show that our proposed framework outperforms many high-ranking background subtraction models. Our contributions can be summarized as follows:Inspired by the filling of missing pixels via the inpainting technique, we adopt a video completion module for modeling the background scene. In order to generate a clean background frame, foreground objects will be substituted by the estimated background colors. Guided by the optical flow, the video completion module can generate good background model for video captured by moving camera, which is not possible for other existing methods.We adopt the BSUV-Net 2.0 [13] for background/foreground segmentation. Although the model is pre-trained with the CDNet [14] video dataset, it can also segment the foreground in unseen videos. However, most of the videos in CDNet are captured by static camera. Although BSUV-Net 2.0 is enhanced with more training videos of moving camera, we find that the model still produces many FP and FN errors. Therefore, we replace the background frame generation method of BSUV-Net 2.0 with our video completion-based background modeler.We propose a framework that comprises video completion-based background modeler and the enhanced BSUV-Net 2.0 foreground segmentation network. To thoroughly evaluate the new framework, we create our own video dataset with videos captured by PTZ camera and free-moving camera. The results show that our framework outperforms many high-ranking background subtraction models.

The paper is organized as follows. The related research studies on background subtraction, particularly with videos captured by moving camera, are reviewed in the following section. Section 3 elaborates our saliency detection framework. We compare our framework with other high-ranking background subtraction algorithms. Quantitative and visual results are presented in Section 4. Discussion is also made on the performance of all these methods. Finally, we draw the conclusion in Section 5.

## 2. Related Work

Many methods have been proposed for segmenting foreground in videos captured by stationary cameras. In this section, we review sophisticated methods that are proposed to handle videos captured by moving cameras. Moving cameras can be categorized into two types: constrained moving camera and freely moving camera. For instance, PTZ camera belongs to the first category. In the second category, examples are hand-held camera, smartphone or camera mounted on drones. Methods developed for constrained camera may not perform well with freely moving camera.

Hishinuma et al. [15] considered the camera small pan/tilt motion as translational. The translation amount is computed from the correlation of the FFT phase terms of stationary background blocks. The synthesized still background model is then used for foreground segmentation. In [16], camera motion is compensated by calculating the homography transformation between two image frames. Scene model, which is a panoramic background, is then generated from the motion compensated video. Foreground objects are detected by comparing the panoramic background with individual image frames of the video. Szolgay et al. [4] proposed a method for detecting moving objects in video taken by a wearable camera. Global camera motion is estimated first by a hierarchical block matching algorithm and then refined by a robust motion estimator. The foreground is identified as the difference between motion-compensated image frames. Tao and Ling [17] proposed a neural network for segmenting foreground in videos captured by PTZ cameras. Deep learning features are extracted by a pre-trained network. Homography matrix is estimated from previous image frames and current image frame with a semantic attention based deep homography estimator. The warped previous frames, current frames, and their features are fed into the fusion network for foreground mask prediction. Komagal and Yogameena [18] reviewed the methods and the datasets for foreground segmentation research with PTZ camera.

With the use of a freely moving camera, both background and foreground are changing. The assumption of background modeling may be violated. For instance, when background and foreground motions are similar, the background model is contaminated with foreground colors. In another scenario, inaccurate camera motion estimation will give rise to false positive errors. Yun et al. [19] proposed an adaptive scheme that can update the background model in accordance with the changes of background. The scheme compensates three types of change—background motion produced by moving camera, foreground motion, and illumination change. Knowing that an explicit camera motion model is not reliable, Sajid et al. [20] proposed an online framework such that both background and foreground models are continuously updated. Background motion is estimated with a low-rank approximation. Motion and appearance models are combined to produce the background/foreground classification. Zhu and Elgammal [21] proposed a multi-layered framework for background subtraction. In each layer, both motion and appearance model are estimated and used for foreground detection. A probability map is inferred by kernel density estimator [5]. Finally, a segmented foreground is generated from the multi-layered outputs by multi-label graph-cut. Chapel and Bouwmans [22] reviewed moving object detection methods with moving camera. They grouped methods into two categories in accordance with scene representation—single-plane and multi-plane. Methods in the first group may generate a panoramic background by image mosaics. Some methods detect moving objects via motion segmentation. Multi-plane approach estimates several planes (may be real or not) as scene representation. Matched feature points are located and eventually used for background/foreground classification.

We adopt the background-centric approach for saliency detection. Instead of modeling the background based on camera motion compensation, which may be inaccurate, we generate and update the background dynamically via video completion and continuous monitoring of foreground segmentation result. Our background modeler, based on the optical flow information, can generate a much better background frame for video captured by a moving camera than other methods. As demonstrated in our results, our framework with the cascade of background modeler and deep-learning foreground segmenter outperforms many high-ranking background subtraction models in saliency detection.

Various video datasets were created for background subtraction research. The CDNet 2014 dataset [14] contains videos grouped under 11 categories. Each video record provides the original image sequence and the corresponding ground truths. Many of the videos were captured in different challenging scenes. For instance, the “PTZ” category contains four videos captured by PTZ camera. The Hopkins 155 dataset [23] contains indoor and outdoor panning videos. Perazzi et al. [24] proposed three versions of the Densely Annotated Video Segmentation (DAVIS) dataset. Some videos were captured by shaking camera. The SegTrack v2 dataset [25] contains videos captured by a moving camera with ground truth of the moving object. The Labeled and Annotated Sequences for Integral Evaluation of SegmenTation Algorithms (LASIESTA) [26] database contains real indoor and outdoor videos with pan, tilt, or shaking cameras. In our experimentations, we create our own dataset comprising videos captured by PTZ camera and moving camera extracted from various publicly available video datasets.

## 3. Saliency Detection Framework

The saliency detection framework SD-BMC is shown in Figure 1. It performs two main tasks—generation of initial background model and continuous saliency detection with updating of background model. First, in part (a), we initialize the system with the first 100 frames. We use the background image generated by the foreground segmenter (BSUV-Net 2.0 [13]) with median filter to create masks in this step. These masks, together with the initial image sequence, will be placed into the video completion-based background modeler (FGVC [27]). From the sequence of completed frames, the most recent one is selected as the background frame (see the dotted arrow). In the stream of saliency detection in part (b), the initial background frame and the current image sequence will be input to the foreground segmenter. The foreground segmenter is pre-trained with CDNet video dataset. As shown in our results, it also performs well with unseen videos. Therefore, we do not retrain the foreground segmenter. The background model will be updated based on the new input masks that are the current foreground segmentation results. The stream of saliency detection with feedback will continue until all video frames are processed.

### 3.1. Evaluation Metrics

In order to evaluate the performance of our framework, we compute eight quantitative measures: *Recall*, *Specificity*, False Positive Rate (*FPR*), False Negative Rate (*FNR*), Percentage of Wrong Classifications (*PWC*), *F-Measure*, *Precision*, and Matthew’s Correlation Coefficient (*MCC*) (where *TP* is true positive, *FP* is false positive, *FN* is false negative, and *TN* is true negative). *F-Measure*, calculated based on *Recall* and *Precision*, is often used as a single numeric measure for ranking different methods.
(1)$$Recall=\frac{TP}{TP+FN}$$
(2)$$Specificity=\frac{TN}{TN+FP}$$
(3)$$FPR=\frac{FP}{FP+TN}$$
(4)$$FNR=\frac{FN}{TP+FN}$$
(5)$$PWC=\frac{100\times \left(FN+FP\right)}{TP+FN+FP+TN}$$
(6)$$F-Measure=\frac{2\times Precision\times Recall}{Precision+Recall}$$
(7)$$Precision=\frac{TP}{TP+FP}$$
(8)$$MCC=\frac{TP\times TN-FP\times FN}{\sqrt{\left(TP+FP\right)\left(TP+FN\right)\left(TN+FP\right)\left(TN+FN\right)}}$$

### 3.2. Background Modeler

Many background modeling algorithms can estimate a clean background frame and even the image sequence contains moving objects. However, if the foreground objects exist too long, there will be phenomena such as ghosts in the background image. The problem becomes more complicated with video captured by a moving camera. Deep learning-based methods have been proposed for background modeling. For instance, Farnoosh et al. [28] proposed a variational autoencoder (VAE) framework for background estimation from videos recorded by fixed camera. In our experimentation on videos of moving camera, there are always blur pixels existing in the final background images.

We adopt and modify the video completion method FGVC [27] for background modeling. The algorithm can generate a clean background image with more attention to the masks corresponding to the foreground objects and also the changing scene between adjacent image frames. Figure 2 shows our video completion-based background modeler. In part (a), the color video sequence and the corresponding binary masks are input to the background modeler. The masks are the foreground regions that need to be completed. They are the results of foreground segmenter, as shown in Figure 1b. Next, in part (b), optical flow *F* between adjacent frames is computed with FlowNet2 [29]. The forward flow is computed from *I_i_* to *I_i_*_+1_.
(9)$$F_{i\to i+1}=Ӻ\left(I_{i},I_{i+1}\right)$$

The backward flow is computed from *I_i_*_+1_ to *I_i_*.
(10)$$F_{i+1\to i}=Ӻ\left(I_{i+1},I_{i}\right)$$

Moreover, flow between some non-adjacent frames is also computed. This can help to estimate missing background colors when camera motion is large. The backward flow and forward flow are predicted from the color video sequence and the corresponding masks. In each flow field, flow edges are extracted. Guided by the flow edge map, a completed optical flow field is generated. In part (c), a set of candidate pixels are computed for each missing pixel. Most of the missing pixels can be filled with inpainting via fusion of the candidate pixels. After that, the network will use Poisson reconstruction [30] to generate the initial completed background frame:(11)$$\underset{\tilde{I}}{\mathrm{argmin}}\|\Delta_{x}\tilde{I}-{\tilde{G}}_{x}{\|}_{2}^{2}+\|\Delta_{y}\tilde{I}-{\tilde{G}}_{y}{\|}_{2}^{2}$$
where $\tilde{I}$ is the weighted average image, and $\tilde{G}$ is the weighted average gradient. Finally, in the last part (d), the modeler will fix the remaining missing pixels with a number of inpainting iterations until there is no missing pixel.

Experimentation is performed to determine the length of the video sequence for background generation. If it is too short, there may not be a sufficient number of candidate pixels for background color synthesis. If it is too long, the time for background generation will be long, and the actual saliency detection will be delayed. First, we choose 30 frames for background generation. Figure 3 shows background modeling on two videos. It can be observed that ghosts exist in the background frame. Then, we lengthen the initialization sequence to 100 frames. Most of the foreground pixels can be substituted with background colors. Table 1 compares the F-measure of saliency detection on the PTZ category of CDNet 2014 dataset. Finally, we fix the length to 100 frames.

### 3.3. Foreground Segmentation

We adopt BSUV-Net 2.0 [13] as foreground segmenter. As shown in Figure 4, it has a U-Net [31] like structure. Based on BSUV-Net [32], BSUV-Net 2.0 further improves background subtraction performance on complicated videos with more spatio-temporal data augmentations. The encoder–decoder structure contains five convolutional blocks in the downsampling path, four convolutional blocks in the upsampling path, and their links via concatenation. The detail of the configuration is shown in Table 2.

An empty background frame, a recent background frame, the current frame and corresponding foreground probability maps (FPM) are needed for background/foreground separation. The input has a total of 12 channels. In order to avoid overfitting problems and to increase the generalization of the network, it uses a batch normalization layer for each convolution layer in the encoder part and a convolution transpose layer in the decoder part. Spatial dropout layers are also used before max-pooling to make the network more generative. Finally, the network uses the sigmoid function to obtain the prediction value *S*(*x*) of the pixels in output binary saliency detection:(12)$$S\left(x\right)=\frac{1}{1+e^{-x}}$$
where *x* is the output of a neuron in the last layer.

Tezcan et al. [13] simulated some changes, e.g., changes that look similar to videos captured by PTZ camera, for data augmentation in training the model. However, as shown in our experimental results, BSUV-Net 2.0 is still not good enough in saliency detection with videos captured by PTZ camera and freely moving cameras. It is because the background modeling method cannot generate a fairly good background frame for complicated videos. Therefore, in our saliency detection stream, we disable the default background modeling method. Instead, we use video completion-based background modeler that can generate a better background frame.

## 4. Result and Discussion

### 4.1. Datasets

We test our saliency detection framework on CDNet 2014 dataset [14] and our customized dataset. As shown in Table 3, CDNet 2014 comprises 11 categories, each of which contains four to six videos. Each video record provides the original image sequence and the corresponding ground truths. Some videos, e.g., the PTZ category, were captured in challenging scenes.

Our customized dataset comprises 22 videos from the FBMS dataset [33] and eight videos from the LASIESTA dataset [26]. The videos were captured by handheld cameras and PTZ cameras. For videos selected from the FBMS dataset, we used manually defined ground truths for 20 continuous frames randomly chosen after the 100th frame in each video. For videos from the LASIESTA dataset, each record provides a number of ground truth images. Table 4 shows the details of our customized dataset.

### 4.2. Performance Evaluation

We implement SD-BMC with the Python-based Pytorch. The computing platform comprised Intel Xeon Silver 4108 CPU 1.8 G 16 Cores and a HPC Cluster with NVIDIA RTX 2080 Ti 11 GB × 8 GPU nodes. The background frame, either in the initialization or in the updating process, is generated from a sequence of 100 image frames. Therefore, each video is partitioned into sections of 100 frames. If the last section contains less than 100 frames, we input all remaining frames into our framework. We resized the original image sequence and ground truth images with a resolution of 320 × 240.

We compare SD-BMC with six background subtraction algorithms—BSUV-Net [32], BSUV-Net 2.0 [13], Fast BSUV-Net 2.0 [13], PAWCS [34], SuBSENSE [35], and ViBe [36]. Tezcan et al. [32] first proposed the BSUV-Net. Background frames were estimated from the video. The current frame of the video and the background frames are input into the fully convolutional neural network for background subtraction. They proposed the second version of the model [13] by training with data simulating spatiotemporal changes. Moreover, they developed the Fast BSUV-Net 2.0 [13] which is a real-time version of the model. St-Charles et al. proposed SuBSENSE [35] and PAWCS [34] for change detection. The background model is a codebook that is generated based on the persistence of pixel features. They are among the high-ranking methods in CDNet 2014. Barnich et al. [36] adopted the bag-of-words approach and proposed an efficient background subtraction method ViBe. At each pixel location, some samples are randomly selected from the image sequence and stored as background colors. The background model is also updated with a random process.

### 4.3. Quantitative and Visual Results

Table 5 shows the numerical results of SD-BMC on CDNet 2014 dataset. Table 6 shows the average results of BSUV-Net, BSUV-Net 2.0, Fast BSUV-Net 2.0, and SD-BMC. The best results are highlighted in red. The second best results are highlighted in blue. Table 7 compares the results of BSUV-Net, BSUV-Net 2.0, Fast BSUV-Net 2.0, and SD-BMC on the PTZ category of CDNet 2014. Figure 5 shows some visual results of BSUV-Net 2.0 and SD-BMC.

Our saliency detection framework achieves the best average Recall, FPR, PWC, and Precision on CDNet 2014 dataset. As shown in the visual results, SD-BMC can achieve comparable performance as BSUV-Net 2.0 in many video categories. The superiority of SD-BMC over BSUV-Net 2.0 can be observed in the PTZ category. Due to the poor background frame, BSUV-Net 2.0 produces a large amount of FP errors. The quantitative results shown in Table 7 clearly indicate that SD-BMC outperforms the other three models in all evaluation metrics on PTZ videos. When a single numeric result, F-measure, is chosen for ranking, SD-BMC outperforms all other methods by more than 11%.

Table 8 shows the average results of BSUV-Net, BSUV-Net 2.0, PAWCS, SuBSENSE, ViBe, and SD-BMC on customized dataset. Table 9 compares the F-measure on individual videos. Table 10 compares the MCC on individual videos. Figure 6 shows some visual results of BSUV-Net, BSUV-Net 2.0, PAWCS, SuBSENSE, ViBe, and SD-BMC.

The videos in the customized dataset are more challenging. We classify the videos, in accordance to their contents, into three groups: animals, people, and things. SD-BMC achieves the best average Recall, FNR, PWC, MCC, and F-Measure. We select a single numeric result, F-measure, for assessing the performance of all methods on individual videos. As shown in Table 9, SD-BMC achieves the best F-measure in many videos. The average F-measure in “animals” and “things” groups are higher than all other methods, while in “people” group, the average F-measure is slightly lower than BSUV-Net 2.0. Similarly, as shown in Table 10, SD-BMC achieves the best average MCC in “animals” and “things” groups, while in “people” group the average MCC is second best. Moreover, all MCC results of SD-BMC are positive, while other methods have some negative MCC due to large FP and FN errors. As shown in Figure 6, SD-BMC can detect saliency very close to the ground truth. BSUV-Net, PAWCS, SuBSENSE, and ViBe produce many FP errors. The second best method, BSUV-Net 2.0, produces more FP and FN errors than SD-BMC. As shown in video marple3, BSUV-Net 2.0 produces FN errors due to camouflage. SD-BMC can segment the human with fewer FN errors. Overall, SD-BMC outperforms BSUV-Net 2.0 in F-Measure result by more than 3%.

### 4.4. Comparative Analysis

According to the results on the CDNet 2014 dataset, we found that SD-BMC outperforms other methods in the PTZ category. For other video categories, SD-BMC achieves comparable results with other methods. The reason is that our video completion-based background modeler, together with the feedback scheme, can generate clear and updated background images. FGVC is a non-scene-specific method, which could be generalized to unseen videos. As FGVC can capture temporal and spatial information, this modeler can generate much better background images. On contrary, the empty background images used in BSUV-Net 2.0 are very blur, which could significantly affect the final saliency detection result. PAWCS and SuBSENSE, which are designed for fixed camera, produce even worse background frame. Figure 7 shows the comparison of background images.

In Table 9, the F-Measure values of BSUV-Net and PAWCS on “cats06” video are left blank. It is because these two methods have zero Recall value. Therefore, their F-Measures cannot be calculated. It is clear that methods that are designed for fixed camera, e.g., PAWCS, SuBSENSE, and ViBe, cannot achieve accurate background subtraction on videos captured by moving camera. The deep learning model BSUV-Net, which can tackle unseen videos, also fails to detect the foreground in many videos of the customized dataset. The enhanced model BSUV-Net 2.0, which is trained with PTZ such as augmented data, performs much better than the first version. SD-BMC, as compared with BSUV-Net 2.0, can detect more accurate saliency with fewer FP and FN errors.

## 5. Conclusions

We propose a new framework, SD-BMC, for the detection of salient regions in each video frames. The framework contains two major modules: video completion-based background modeler and the deep learning-based foreground segmenter network. In order to enable our framework for long-term saliency detection, the background modeler can adjust the background image dynamically via a feedback mechanism. SD-BMC can best segment foregrounds in videos captured by moving camera. In order to demonstrate this capability, we create our customized dataset with challenging videos captured by PTZ camera and handheld camera. The results, obtained from the PTZ videos, show that our proposed framework outperforms some deep learning-based background subtraction models by 11% or more in the F-Measure result. With more challenging videos, our framework also outperforms many high-ranking background subtraction methods by more than 3%.

Although the results show that SD-BMC is superior to other deterministic as well as deep learning-based background subtraction methods, there are still more method that can further improve it. In this study, we focus on designing a saliency detection framework for moving camera videos. In the future, we will work on new models for other challenging scenarios. The background modeling process can be made faster in order to tackle abrupt changes. Moreover, the foreground segmenter can adopt a teacher–student structure. While the complex teacher model is used in the training process, the testing process will be performed by a simpler student model. The lite model can be used for real-time saliency detection.

## Figures and Tables

**Figure 1 sensors-21-08374-f001:**
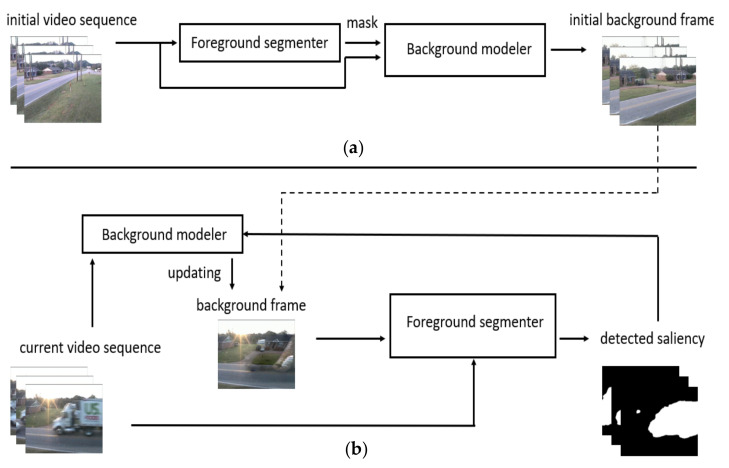
Overview of the saliency detection framework: (**a**) background model initialization; (**b**) continuous saliency detection.

**Figure 2 sensors-21-08374-f002:**
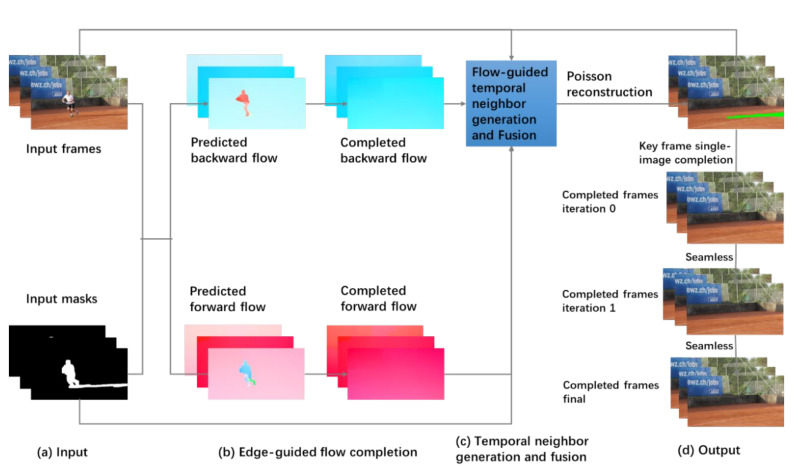
Video completion-based background modeler.

**Figure 3 sensors-21-08374-f003:**
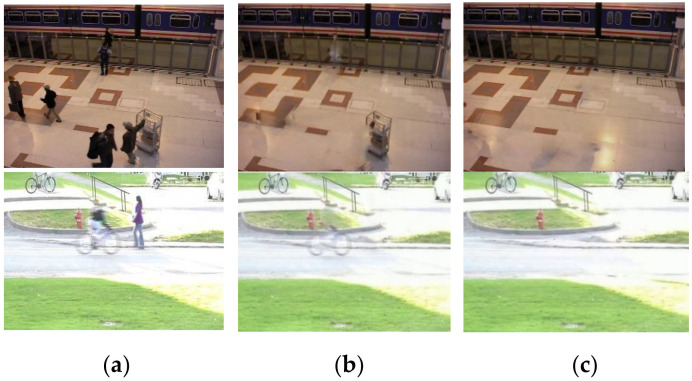
Visual results of background modeling: (**a**) original frame; (**b**) 30 initialization frames; and (**c**) 100 initialization frames.

**Figure 4 sensors-21-08374-f004:**
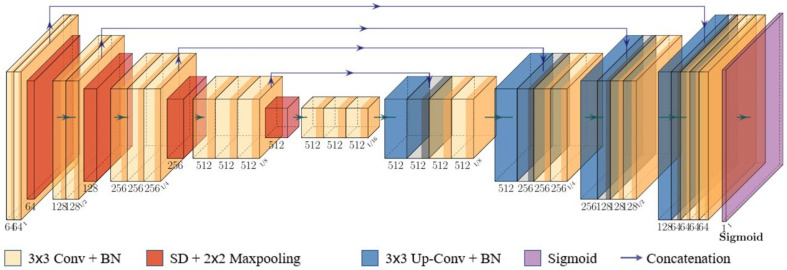
Structure of foreground segmenter.

**Figure 5 sensors-21-08374-f005:**
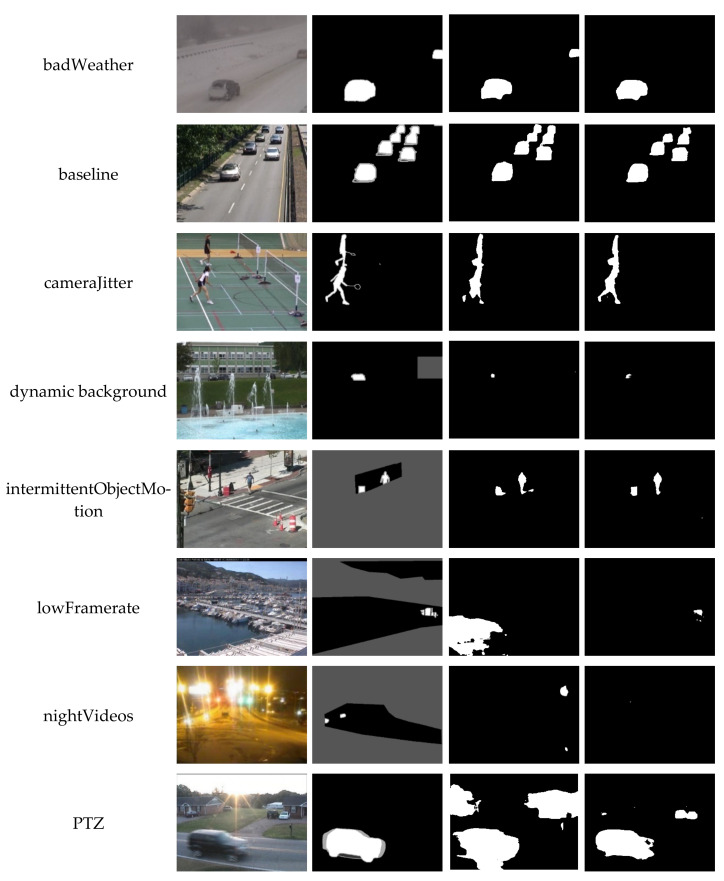
Visual results of BSUV-Net 2.0 and SD-BMC on CDNet 2014.

**Figure 6 sensors-21-08374-f006:**
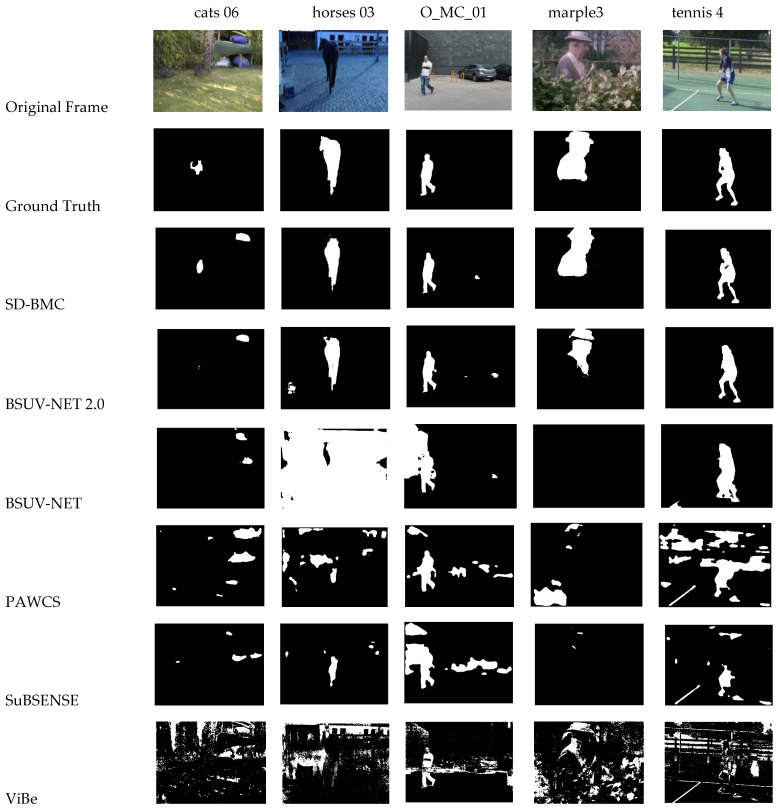
Visual results on customized dataset.

**Figure 7 sensors-21-08374-f007:**
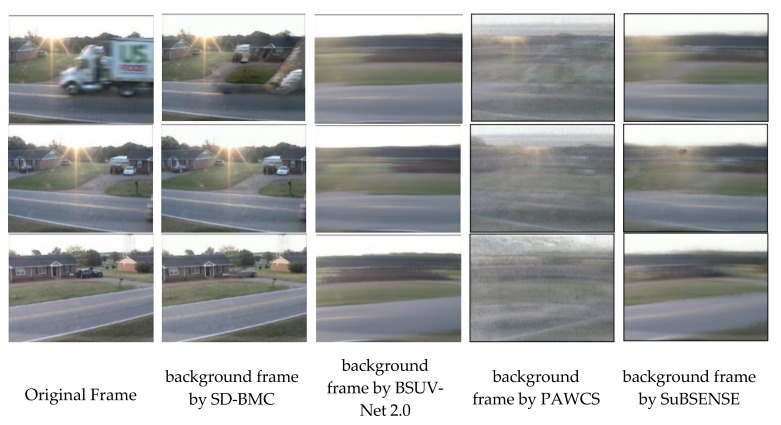
Comparison of background frames used in BSUV-Net 2.0, PAWCS, SuBSENSE, and SD-BMC on PTZ video.

**Table 1 sensors-21-08374-t001:** F-measure of saliency detection with two settings for background modeling.

Length of Initialization Sequence	F-Measure
30 frames	0.8062
100 frames	**0.8147**

**Table 2 sensors-21-08374-t002:** Layer Configuration of Foreground Segmenter (SD: spatial dropout layer; BN: batch normalization).

Network Structure of BSUV	Kernel Size	In_Channels	Out_Channels
Conv + BN	3 * 3	12	64
Conv + BN	3 * 3	64	64
SD + Maxpooling	2 * 2	64	64
Conv + BN	3 * 3	64	128
Conv + BN	3 * 3	128	128
SD + Maxpooling	2 * 2	128	128
Conv + BN	3 * 3	128	256
Conv + BN	3 * 3	256	256
Conv + BN	3 * 3	256	256
SD + Maxpooling	2 * 2	256	256
Conv + BN	3 * 3	256	512
Conv + BN	3 * 3	512	512
Conv + BN	3 * 3	512	512
SD + Maxpooling	2 * 2	512	512
Conv + BN	3 * 3	512	512
Conv + BN	3 * 3	512	512
Conv + BN	3 * 3	512	512
Up-Conv + BN	3 * 3	512	512
Conv + BN	3 * 3	512 + 512	512
Conv + BN	3 * 3	512	512
Up-Conv + BN	3 * 3	512	512
Conv + BN	3 * 3	512 + 256	256
Conv + BN	3 * 3	256	256
Up-Conv + BN	3 * 3	256	256
Conv + BN	3 * 3	256 + 128	128
Conv + BN	3 * 3	128	128
Up-Conv + BN	3 * 3	128	128
Conv + BN	3 * 3	128 + 64	64
Conv + BN	3 * 3	64	64
Conv + BN	3 * 3	64	1
Sigmoid		1	1

**Table 3 sensors-21-08374-t003:** CDNet 2014 Dataset Categories and Video Scene.

Category	Video Scenes					
Bad Weather	Blizzard (7000 Frames)	Skating (3900 Frames)	SnowFall (6500 Frames)	WetSnow (3500 Frames)		
Low Framerate	port 0 17fps (3000 frames)	tramCrossroad 1fps (900 frames)	tunnelExit 0 35fps (4000 frames)	turnpike 0 5fps (1500 frames)		
Night Videos	bridgeEntry (2500 frames)	busyBoulvard (2760 frames)	fluidHighway (1364 frames)	streetCornerAtNight (5200 frames)	tramStation (3000 frames)	winterStreet (1785 frames)
PTZ	continuousPan (1700 frames)	intermittentPan (3500 frames)	twoPositionPTZCam (2300 frames)	zoonInZoomOut (1130 frames)		
Thermal	corridor (5400 frames)	library (4900 frames)	park (600 frames)	diningRoom (3700 frames)	lakeSide (6500 frames)	
Shadow	backdoor (2000 frames)	bungalows (1700 frames)	busStation (1250 frames)	cubicle (7400 frames)	peopleInShade (1199 frames)	copyMachine (3400 frames)
Intermittent Object Motion	abandonedBox (4500 frames)	parking (2500 frames)	streetLight (3200 frames)	sofa (2750 frames)	tramstop (3200 frames)	winterDriveway (2500 frames)
Camera Jitter	badminton (1150 frames)	boulevard (2500 frames)	sidewalk (1200 frames)	traffic (1570 frames)		
Dynamic Background	boats (7999 frames)	canoe (1189 frames)	fountain01 (1184 frames)	fountain02 (1499 frames)	overpass (3000 frames)	fall (4000 frames)
Baseline	highway (1700 frames)	office (2050 frames)	pedestrians (1099 frames)	PETS2006 (1200 frames)		
Turbulence	turbulence0 (5000 frames)	turbulence1 (4000 frames)	turbulence2 (4500 frames)	turbulence3 (2200 frames)		

**Table 4 sensors-21-08374-t004:** Customized Dataset Categories and Video Scene.

Category	Video	Number of Frames	Ground Truth Frames
**animals**	cats06	331	(190, 209)
	cats07	193	(110, 129)
	dogs02	420	(160, 179)
	horses01	500	(280, 299)
	horses02	240	(140, 159)
	horses03	240	(214, 233)
	horses04	800	(540, 559)
	horses05	456	(320, 339)
	horses06	720	(380, 399)
	rabbits01	290	(200, 219)
**people**	I_MC_01	300	(1, 300)
	I_SM_01	300	(1, 300)
	I_SM_02	300	(1, 300)
	I_SM_03	300	(1, 300)
	marple1	328	(309, 328)
	marple2	250	(220, 239)
	marple3	323	(250, 269)
	marple6	800	(180, 199)
	marple7	528	(370, 389)
	marple10	460	(440, 459)
	marple11	173	(150, 169)
	O_MC_01	425	(1, 425)
	O_SM_01	425	(1, 425)
	O_SM_02	425	(1, 425)
	O_SM_03	425	(1, 425)
	people03	180	(160, 179)
	people04	320	(270, 289)
	people05	260	(140, 159)
**things**	farm01	252	(194, 213)
	tennis	466	(281, 300)

**Table 5 sensors-21-08374-t005:** Evaluation Metrics of SD-BMC on CDNet 2014.

Category\Metric	*Recall*	*Specificity*	*FPR*	*FNR*	*PWC*	*F-Measure*	*Precision*
**PTZ**	0.8798	0.9970	0.0030	0.1202	0.3788	0.8147	0.7880
**badWeather**	0.7139	0.9997	0.0003	0.2861	0.4538	0.8155	0.9790
**baseline**	0.9325	0.9987	0.0013	0.0675	0.2642	0.9514	0.9736
**cameraJitter**	0.8310	0.9973	0.0027	0.1690	0.9369	0.8790	0.9382
**dynamic background**	0.7437	0.9998	0.0002	0.2563	0.1229	0.8013	0.9336
**intermittentObjectMotion**	0.7987	0.9980	0.0020	0.2013	1.3093	0.8758	0.9809
**lowFramerate**	0.5614	0.9994	0.0006	0.4386	0.6950	0.6292	0.9046
**nightVideos**	0.4856	0.9992	0.0008	0.5144	1.1114	0.5727	0.9328
**shadow**	0.9389	0.9984	0.0016	0.0611	0.4378	0.9543	0.9704
**thermal**	0.6156	0.9990	0.0010	0.3844	1.2019	0.7075	0.9531
**turbulence**	0.5732	0.9998	0.0002	0.4268	0.2583	0.7107	0.9654
**Average value**	0.7340	0.9988	0.0012	0.2660	0.6518	0.7920	0.9381

**Table 6 sensors-21-08374-t006:** Average evaluation metrics of BSUV-Net, BSUV-Net 2.0, Fast BSUV-Net 2.0, and SD-BMC on CDNet 2014.

Method	*Recall*	*Specificity*	*FPR*	*FNR*	*PWC*	*F-Measure*	*Precision*
BSUV-Net	0.8203	0.9946	0.0054	0.1797	1.1402	0.7868	0.8113
BSUV-Net 2.0	0.8136	0.9979	0.0021	0.1864	0.7614	0.8387	0.9011
Fast BSUV-Net 2.0	0.8181	0.9956	0.0044	0.1819	0.9054	0.8039	0.8425
SD-BMC	0.7340	0.9988	0.0012	0.2660	0.6518	0.7920	0.9381

**Table 7 sensors-21-08374-t007:** Evaluation metrics of BSUV-Net, BSUV-Net 2.0, Fast BSUV-Net 2.0, and SD-BMC on PTZ category of CDNet 2014.

Method	*Recall*	*Specificity*	*FPR*	*FNR*	*PWC*	*F-Measure*	*Precision*
BSUV-Net	0.8045	0.9909	0.0091	0.1955	1.0716	0.6282	0.5897
BSUV-Net 2.0	0.7932	0.9957	0.0043	0.2068	0.5892	0.7037	0.6829
Fast BSUV-Net 2.0	0.8056	0.9878	0.0122	0.1944	1.3516	0.5014	0.4236
SD-BMC	0.8798	0.9970	0.0030	0.1202	0.3788	0.8147	0.7880

**Table 8 sensors-21-08374-t008:** Average evaluation metrics of BSUV-Net, BSUV-Net 2.0, PAWCS, SuBSENSE, ViBe, and SD-BMC on customized dataset.

Method	*Recall*	*Specificity*	*FPR*	*FNR*	*PWC*	*F-Measure*	*Precision*
BSUV-Net	0.4823	0.8556	0.1444	0.5177	18.4314	0.3707	0.4547
BSUV-Net 2.0	0.5273	0.9934	0.0066	0.4727	7.3303	0.6123	0.8827
PAWCS	0.4730	0.8695	0.1305	0.5270	19.0191	0.3323	0.3055
SuBSENSE	0.3066	0.9460	0.0540	0.6934	13.9365	0.2936	0.4167
ViBe	0.1332	0.7947	0.2053	0.6680	25.4062	0.2001	0.1707
SD-BMC	0.5635	0.9906	0.0094	0.4365	7.1580	0.6446	0.8719

**Table 9 sensors-21-08374-t009:** F-measure of BSUV-Net, BSUV-Net 2.0, PAWCS, SuBSENSE, ViBe, and SD-BMC on Individual Videos of the Customized Dataset.

F-Measure Comparison on Individual Video	Video	SD-BMC	BSUV-Net 2.0	BSUV-Net	PAWCS	SuBSENSE	ViBe
**animals**	cats06	0.4417	0.0041			0.0005	0.0062
	cats07	0.5700	0.6634	0.5549	0.7506	0.4268	0.4408
	dogs02	0.7758	0.6531	0.1754	0.4514	0.6144	0.0981
	horses01	0.7163	0.7075	0.5733	0.5131	0.7398	0.2828
	horses02	0.3591	0.4368	0.3474	0.1090	0.0264	0.0485
	horses03	0.8341	0.8130	0.1235	0.1844	0.2957	0.0454
	horses04	0.2322	0.2198	0.4269	0.0018	0.2025	0.0449
	horses05	0.1433	0.1493	0.6209	0.4218	0.1793	0.6971
	horses06	0.4626	0.4583	0.1986	0.2434	0.4240	0.0353
	rabbits01	0.1972	0.1607	0.1805	0.2484	0.1214	0.0239
	**Average**	0.4732	0.4266	0.3557	0.3249	0.3031	0.1723
**people**	I_MC_01	0.8343	0.8495	0.6551	0.3541	0.4782	0.1049
	I_SM_01	0.7957	0.8631	0.2943	0.4522	0.6024	0.2066
	I_SM_02	0.7577	0.8310	0.3354	0.4601	0.5193	0.2312
	I_SM_03	0.7283	0.8248	0.3593	0.4367	0.4462	0.2334
	marple1	0.6106	0.6042	0.3784	0.2276	0.3022	0.0946
	marple2	0.8292	0.9490	0.0586	0.0312	0.0085	0.4303
	marple3	0.9186	0.4303	0.0013	0.0661	0.0050	0.2659
	marple6	0.4521	0.3793	0.2967	0.3744	0.4677	0.3463
	marple7	0.9308	0.8645	0.5030	0.0032	0.0452	0.0895
	marple10	0.6870	0.8286	0.0016	0.0429	0.0443	0.0540
	marple11	0.4359	0.7044	0.4371	0.1585	0.0344	0.3260
	O_MC_01	0.7097	0.6292	0.2610	0.3230	0.2195	0.1570
	O_SM_01	0.8686	0.8704	0.3575	0.3559	0.2926	0.1446
	O_SM_02	0.8113	0.8298	0.1920	0.3108	0.3205	0.1067
	O_SM_03	0.7955	0.8115	0.1516	0.2569	0.2894	0.0909
	people03	0.5481	0.4985	0.2137	0.4588	0.0144	0.2158
	people04	0.8333	0.8457	0.3818	0.1923	0.2498	0.0628
	people05	0.8786	0.8662	0.6771	0.7264	0.6623	0.3420
	**Average**	0.7459	0.7489	0.3086	0.2906	0.2779	0.1946
**things**	farm01	0.5936	0.4329	0.1859	0.3668	0.1654	0.2630
	tennis	0.8360	0.8902	0.6442	0.2836	0.4341	0.2035
	**Average**	0.7148	0.6616	0.4151	0.3252	0.2998	0.2333
**Total Average**		0.6446	0.6123	0.3707	0.3323	0.2936	0.2001

**Table 10 sensors-21-08374-t010:** MCC of BSUV-Net, BSUV-Net 2.0, PAWCS, SuBSENSE, ViBe, and SD-BMC on Individual Videos of the Customized Dataset.

MCC Comparison on Individual Video	Video	SD-BMC	BSUV-Net 2.0	BSUV-Net	PAWCS	SuBSENSE	ViBe
**animals**	cats06	0.4527	−0.0054	−0.0120	−0.0264	−0.0134	−0.0242
	cats07	0.6181	0.6922	0.6064	0.7597	0.4867	0.4224
	dogs02	0.7837	0.6833	0.1643	0.4546	0.6059	0.0826
	horses01	0.7145	0.7071	0.5188	0.4147	0.7110	0.0906
	horses02	0.4261	0.5021	0.3021	0.0505	0.0068	−0.0978
	horses03	0.8260	0.8008	−0.2155	0.1212	0.3344	−0.0733
	horses04	0.3097	0.3064	0.2132	0.0203	0.2379	−0.1266
	horses05	0.1890	0.1931	−0.0718	0.3369	0.1708	0.5011
	horses06	0.5152	0.5165	0.1678	0.2073	0.3970	0.0060
	rabbits01	0.1594	0.1207	0.2417	0.2900	0.0959	−0.0811
	**Average**	** 0.4994 **	** 0.4517 **	**0.1915**	**0.2629**	**0.3033**	**0.0700**
**people**	I_MC_01	0.8370	0.8552	0.6597	0.3794	0.4837	0.1186
	I_SM_01	0.7801	0.8531	0.2879	0.4453	0.5758	0.1416
	I_SM_02	0.7304	0.8125	0.2932	0.4315	0.4602	0.1142
	I_SM_03	0.7011	0.8059	0.3226	0.3982	0.3771	0.1186
	marple1	0.5877	0.5836	0.3740	0.0990	0.1064	−0.1778
	marple2	0.7430	0.9223	−0.0360	−0.1704	−0.1508	0.0133
	marple3	0.9090	0.4920	0.0238	−0.0335	0.0461	0.1245
	marple6	0.4511	0.4032	0.2078	0.1964	0.3597	0.0991
	marple7	0.9137	0.8422	0.4403	−0.1902	0.0302	−0.1260
	marple10	0.7114	0.8333	−0.0218	−0.0092	−0.0295	−0.0048
	marple11	0.3744	0.6867	0.4030	0.0470	−0.0467	0.2817
	O_MC_01	0.7071	0.6326	0.3258	0.3975	0.3112	0.2158
	O_SM_01	0.8722	0.8737	0.3973	0.4027	0.3162	0.1670
	O_SM_02	0.8200	0.8365	0.2352	0.3511	0.3420	0.1137
	O_SM_03	0.8056	0.8194	0.1759	0.3017	0.3113	0.0834
	people03	0.5799	0.5420	0.3204	0.4959	0.0778	0.0328
	people04	0.8409	0.8504	0.4327	0.2422	0.2449	0.0123
	people05	0.8648	0.8514	0.6700	0.7042	0.6516	0.2668
	**Average**	** 0.7350 **	** 0.7498 **	**0.3062**	**0.2494**	**0.2482**	**0.0886**
**things**	farm01	0.6131	0.4933	0.2097	0.2273	0.0250	0.0860
	tennis	0.8279	0.8838	0.6233	0.2481	0.3992	0.1327
	**Average**	** 0.7205 **	** 0.6886 **	**0.4165**	**0.2377**	**0.2121**	**0.1094**
**Total Average**		0.6516	0.6300	0.3047	0.2500	0.2545	0.0893

## Data Availability

Not applicable.

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
