# Peer review of "Saliency Detection with Moving Camera via Background Model Completion"

_sensors, 2021, doi:10.3390/s21248374_

Round 1
Reviewer 1 Report
This paper adopts video completion for background modeling and saliency detection in videos captured by moving camera. The experimental results show that the proposed method outperforms the state-of-the-art approaches.
There are some problems of theoretical and experimental analyses in this manuscript and it can be revised in the following aspects.
- In section 3, it is better to supplement necessary mathematical equations.
- In Figure 2, the completed foreground flow should contain objects.
- In Figure 3 (c), the top small car should be removed.
- In Table 2, SD is defined but not utilized.
- In Table 6, 8 and 9, the proposed solution is not superior to the state-of-the-art approaches in all situations. How to explain and improve it?
- In Figure 5, each column should be labeled.
- In Figure 6, the results of the proposed method have false subjects. How explain and improve it?
- In Figure 7, column 2 should be background.
Reviewer 2 Report
The paper is about a method to detect background and foreground in the video.
I'd say a bit more extensive discussion of what is supposed to count as background and foreground (or saliency) here would be beneficial. Is it meant do be application specific? For example, let's say that we see a video of a road and there is a car and a bird. Would they both be counted as foreground? Would they be counted a foreground, if they move relative to background? Or is it the case that a system meant for driving would have to count car as foreground and bird as background, while a system meant for ornithological research would have to count bird as foreground and car as background?
Also, I'd say that it would be a good idea to dedicate a slightly longer section for methods of evaluation of performance of algorithm, for example, "F-measure" (mentioned in section 3.1, defined in section 4.2). For otherwise, for example, Table 1 is left unclear - what is being compared with what? Are the true and detected areas of foreground being compared? For Figure 3 that is just before the table would seem to encourage us to expect something connected to quality of detected background as such...
So, it would be a good idea to explain that you will be dealing with pixels in a plane. it would also be a good idea to point out that, for example, precision and recall are meant to be used together, as are specificity and sensitivity (called "recall" here). That, for example, "F-measure" is meant to be used alone (hard to say, maybe it would be a good idea to add MCC - Matthew's Correlation Coefficient? or would that be too much?).
Also, the formulas in section 4.2. do not look as beautiful, as they could. For example, (6) looks as if it is about difference of two quantities - "F" and "measure", when it is really about one quantity "F-measure". It would be better to use short abbreviations instead of words. For example, "F", "Fm" or "FM" for "F-measure".
Abbreviation "PTZ" should be explained.
Fig. 5 has a caption "Visual results of BSUV-Net 2.0 and SD-BMC on CDNet 2014", but it is not clear. There are four columns. The first column seems to be the frame under consideration (even that has to be guessed). But what do other three columns mean? Which of them shows the result of BSUV-Net 2.0, which of them shows the result of SD-BMC? And what is shown by the one that would be left? The ground truth? Such questions shouldn't arise, they should be answered in the figure or the caption.
The claim in the conclusions that "The results, obtained from the PTZ videos, show that our proposed framework outperforms some deep learning-based background subtraction models by 11% or more." is ambiguous, for many measures were considered (for example, recall, F-measure), and it is not clear which of them is being referred to. Naturally, the same applies to the claim about 3% and those same claims repeated in the abstract.
It is not clear how training was done. In fact, it is not very clear how the system as a whole is supposed to work. Fig. 2 indicates that "Input masks" are used by "Video completion-based background modeler". How are they found? Are they supposed to be made manually? Also, if they are (for each video?), that should be pointed in "Discussion" as a limitation (sure, a method like that can be a useful inspiration for others, but it can't be used for most practical purposes).
Now, strictly speaking, some of such questions might be answered by Fig. 1. But that indicates another problem: the structure of the paper. Fig. 1 is in the "Introduction". But the functioning of the system is explained in section 3., and that is where the reader is going to look for the overview of the system. Furthermore, the "Foreground segmentation" part is explained in section 3.2., but it gives its output to "Background modeler" described in section 3.1., thus the reader should be given at least a hint that an explanation is coming later.
The basic structure of a paper would consist of "Introduction", "Methods", "Results" and "Discussion". I'd say that section 2 ("Related work") could be merged into "Introduction", which is supposed to explain related work anyway, and even does that now (to some extent). Instead, the overview of the system should go to section 3 ("Saliency detection framework", corresponding to "Methods"). Likewise, sections 4.1 to 4.3 seem to describe not results (as name of section 4, "Result and discussion" would indicate), but methods of getting those results. Thus they should go to some sort of "Methods" part.
For that matter, perhaps Fig. 1 (or a different figure) could point out what is being trained (or where weights from somewhere else are used)? For example, the weights from somewhere else could be seen as a different kind of input (to some extent).
Maybe "Poisson reconstruction" (in section 3.1.) should get a reference?
The abstract talks about camouflage ("In another scenario, camouflage will lead to false negative errors."). Did I miss it, or the paper itself does not deal with camouflage (there is the same sentence in "Introduction", but it does not seem to be the most important there, and it seems to lead nowhere important)?
There are some English mistakes, for example, in section 4.1. "For videos selected from the FBMS dataset, we used manually define the ground truths for 20 continuous frames randomly chosen after the 100th frame in each video." should have had "defined" and should have had no "the" afterwards. In the abstract, there is "We adopt the background/foreground segmenter, although pre-trained with a specific video dataset, can also detect saliency in unseen videos.", where the subject seems to be "we", but then "can" doesn't fit it. Was "which" left out after "segmenter"?
Round 2
Reviewer 1 Report
The authors have correctly answered all my questions, the paper has been satisfactorily improved, and it can be considered for further processing.
Reviewer 2 Report
I see that great progress has been made.
In section "Background modeler" it is written "Figure 3 shows the background modeling on two videos.", but frames from three videos are visible (perhaps it is only so because differences are being shown; modified manuscript without marked differences is not available for reviewers).
Near the end of section "Introduction" there is a part that is not really visible because of marking of differences (it is under the mark of removal of Fig. 1). Still, I hope everything is going to be fine there.
I have to say that values of MCC look interesting. Perhaps it is worth to point out in the paper that your method had positive MCC for every single video, while other methods had negative MCC for at least one video.
